# Risk of Burnout among Emergency Department Staff as a Result of Violence and Aggression from Patients and Their Relatives

**DOI:** 10.3390/ijerph19094945

**Published:** 2022-04-19

**Authors:** Anja Schablon, Jan Felix Kersten, Albert Nienhaus, Hans Werner Kottkamp, Wilfried Schnieder, Greta Ullrich, Karin Schäfer, Lisa Ritzenhöfer, Claudia Peters, Tanja Wirth

**Affiliations:** 1Institute for Health Services Research in Dermatology and Nursing, University Medical Center Hamburg-Eppendorf, 20251 Hamburg, Germany; j.kersten@uke.de (J.F.K.); albert.nienhaus@bgw-online.de (A.N.); c.peters@uke.de (C.P.); 2Department of Occupational Medicine, Hazardous Substances and Public Health, Institution for Statutory Accident Insurance and Prevention in the Healthcare and Welfare Services, 22089 Hamburg, Germany; 3Protestant Hospital of the Bethel Foundation, 33617 Bielefeld, Germany; hans-werner.kottkamp@evkb.de; 4Klinikum Herford, Emergency Department, Medizin Campus OWL of the Ruhr University Bochum, 32049 Herford, Germany; wilfried.schnieder@klinikum-herford.de; 5Zentrale Notaufnahme, Paracelsus-Klinik Henstedt-Ulzburg, 24558 Henstedt-Ulzburg, Germany; greta.ullrich@yahoo.de; 6Prevention Service, Institution for Statutory Accident Insurance and Prevention in the Healthcare and Welfare Services, Helmholtzstrasse 2, 80636 Munich, Germany; dr.karin.schaefer@bgw-online.de; 7Prevention Department, Accident Insurance Institution Hessen, Leonardo-da-Vinci-Allee 20, 60486 Frankfurt am Main, Germany; l.ritzenhoefer@ukh.de; 8Institute for Occupational and Maritime Medicine (ZfAM), University Medical Center Hamburg-Eppendorf (UKE), 20459 Hamburg, Germany; t.wirth.ext@uke.de

**Keywords:** workplace violence, emergency department staff, burnout

## Abstract

Emergency department staff are often affected by incidents of violence. The aim of the study was to generate data on the frequency of violence by patients and accompanying relatives and the correlation between experienced aggression, a possible risk of burnout and a high sense of stress. Additionally, the buffering effect of good preventive preparation of care staff by the facility on aggressive visitors and patients was examined. In this cross-sectional study, members of the German Society for Interdisciplinary Emergency and Acute Medicine were surveyed. The investigation of risk factors, particularly experiences of verbal and physical violence, as well as exhaustion and stress, was carried out using ordinal regression models. A total of 349 staff from German emergency departments took part in the survey, 87% of whom had experienced physical violence by patients and 64% by relatives. 97% had been confronted with verbal violence by patients and 94% by relatives. Violence by relatives had a negative effect on perceived stress. High resilience or effective preparation of employees for potential attacks was shown to have a protective effect with regard to the burnout risk and perceived stress. Therefore, management staff play a major role in preventing violence and its impact on employees.

## 1. Introduction

Employees in nursing and care professions frequently experience violence [1,2,3]. The International Labour Organisation (ILO) [4] defines violence in the workplace as “any action, incident or behaviour that departs from reasonable conduct in which a person is assaulted, threatened, harmed, injured in the course of, or as a direct result of, his or her work”. Within the nursing and healthcare sectors, emergency department staff are particularly affected by violent incidents [5,6,7]. Studies have shown that they experienced higher prevalences of physical (31%) and non-physical (62.3%) violence in the past twel- ve months than employees in most other areas of medical care (e.g., primary care 7.1% or general hospital 54.6%) [6]. A meta-analysis calculated the incidence of workplace violence in the emergency department using study data that was primarily based on security recordings and incident reports. For every 10,000 cases in the emergency department, there were on average 36 violent patients (incidence = 0.36%) [8].

A further meta-analysis on experiences of violence in emergency departments included a total of 26 articles, investigating a total of 9072 cases of workplace violence. The overall prevalence of verbal violence was 0.77 (95% confidence interval (CI) 0.72–0.82), indicating that 77% of emergency department staff were affected by workplace violence. 6575 (72%) cases involved verbal violence and 1639 (18%) involved physical violence. The affected emergency department staff consisted of 2112 (36.5%) doctors, 3225 (55.7%) nurses and 455 (7.8%) other emergency department staff [9].

In a study at Frankfurt University Hospital from March 2017 to February 2018, an incidence of 3% was reported (544 incidents for 18,121 treated patients) [10].

Schuffenhauser and Hettmannsperger-Lippolt (2021) examined 250 nurses in a nationwide survey on the prevalence of violence against nurses in emergency departments in Germany. Verbal violence by patients in the last 12 months was reported by 81.1% of female and 74.8% of male participants. The prevalence of physical violence in the past 12 months was 43.9% for women and 37.9% for men. The prevalence of sexualised violence in the last 12 months was 48.7% for women and 29.1% for men. An assosiation between experienced violence and stress was found. This was significant for verbal and physical violence (b = 0.18, *p* = 0.002; b = 0.10, *p* = 0.016) and highly significant (b = 0.11, *p* = 0.001) for sexualised violence. No difference between men and women was observed [11].

The actual figures may be much higher because—as various studies have shown—violent workplace incidents experienced by emergency department staff are systematically under-reported [12,13]. In qualitative studies, employees themselves described workplace violence as “normal” and part of the job [14,15]. Risk factors for violent incidents in emergency departments include, for example, alcohol and drug consumption, patients’ psychiatric/neurological diseases and painful diseases and high numbers of patients with longer waiting times as a result [16,17,18,19]. In addition to physical injuries, violence (such as verbal abuse) may also have serious consequences for the staff with regard to their mental health and wellbeing, as well as posing a higher risk of burnout [3,6,20,21,22,23,24]. A literature review showed that violence against ambulance and emergency room staff is still very insufficiently studied in Germany. At the same time only few violence prevention measures are implemented and, consequently, employees feel inadequately prepared for violent situations [17]. In order to raise awareness on the topic, it is important to obtain knowledge on the occurrence and effects of violence in emergency care in Germany.

The aim of this study is to collect data on the frequency of violent incidents by patients and their relatives directed towards emergency department employees in Germany. Furthermore, the connection between the frequency of aggressive incidents by patients and their relatives, the associated possible risk of burnout and a high stress level of the staff will be investigated. Additionally, the buffering effect of a good preventive preparation of care staff by the facility on aggressive visitors and patients is examined. The role of resilience as a personal resource will also be investigated with regard to the risk of burnout.

### 1.1. Experience of Violence/Aggression and Risk of Burnout and Perceived High Stress

As a result of experiencing violence, emergency department employees described feelings of fear, confusion, anger, depression, guilt, humiliation, helplessness and disappointment [1,16]. The most frequently studied consequences of experiencing violence were a decrease in job satisfaction and an increased risk of burnout. A recent study found that non-physical violence, mainly verbal aggression, was positively associated with emotional exhaustion, cynicism and reduced professional efficacy [25].

In one study from the US, employee exposure to verbal abuse, such as threats made by patients in the emergency department, was significantly associated with burnout, secondary traumatic stress and lower satisfaction with the care provided [22,26,27]. In a study of emergency department employees in Chile, 71% of those surveyed were affected by violence, with 10.5% showing symptoms of burnout. Being a victim of violence was highly associated with emotional exhaustion (Odds Ratio (OR) 1.7, 95% CI 1.1–2.8) and depersonalisation (OR 2.0, 95% CI 1.3–3.3) [28]. The studies cited showed a correlation between violent experiences and a negative impact on mental health, as well as the risk of burnout, in healthcare workers and in emergency departments. However, to our knowledge, there have been no studies with emergency department staff in Germany so far that confirm these associations with burnout risk. In the view of this, we propose the following hypotheses:

**Hypothesis** **1** **(H1).**
*Patient-related physical violence is positively related to work-related burnout (1a), patient-related burnout (1b) and stress perception (1c) of emergency staff.*


**Hypothesis** **2** **(H2).**
*Relative-related physical violence is positively related to work-related burnout (2a), patient-related burnout (2b) and stress perception (2c) of emergency staff.*


**Hypothesis** **3** **(H3).**
*Patient-related verbal aggression is positively related to work-related burnout (3a), patient-related burnout (3b) and stress perception (3c) of emergency staff.*


**Hypothesis** **4** **(H4).**
*Relative-related verbal aggression is positively related to work-related burnout (4a), patient-related burnout (4b) and stress perception (4c) of emergency staff.*


### 1.2. Prevention Measures and Risk of Burnout and Perceived Stress

Preventive measures could help employees feel well prepared for dealing with potential violence from patients or their relatives and thus strengthen the employees’ feeling of security in the workplace. They are categorised as preventive, protective and post-incident approaches. While post-incident approaches aim to reduce the negative impact of violent incidents, preventive and protective approaches aim to reduce the risk of violence or to provide better ways of handling violence [29]. These can be implemented on an environmental, organisational or behavioural level. According to the Guidelines for the Prevention of Workplace Violence in Healthcare, measures such as controlled access to the emergency department, good lighting, comfortable waiting areas, alarm systems and surveillance cameras can improve safety (environmental level). At an organisational level, adequate staffing, open communication about violence and aggression and optimised workflows are important. Potential interventions at the behavioural level include training staff, supervisors and managers with regard to strategies and processes, with de-escalation and self-defence techniques being proven effective [29,30,31,32].

Studies have shown that preventive, protective and post-incident measures can reduce perceived stress and the risk of burnout as a result of violent experiences [1,24,32]. A study on violence and aggression experienced by nursing and care workers in Germany showed that if the workplace prepares employees effectively for potential violent incidents, there is a negative correlation with high levels of perceived stress (OR 0.6, 95% CI 0.4–0.8, *p* = 0.001) [1]. High resilience among those affected may also have a protective effect on wellbeing or with regard to the potential for burnout. Studies showed higher burnout levels among people with lower or moderate resilience [23,33]. Another study emphasised that personal resources such as resilience could play an important role in the context of violence prevention and coping strategies and should be further investigated in order to make an important contribution to recommendations for practice [11]. In this regard, we want to investigate the following hypotheses:

**Hypothesis** **5** **(H5).**
*Where emergency department employees feel well prepared by their workplace for potential violent incidents, there is a negative correlation with work-related burnout (5a), patient-related burnout (5b) and perceived stress (5c) by those affected.*


**Hypothesis** **6** **(H6).**
*Resilience is negatively correlated with work-related burnout (6a) and patient-related burnout (6b) among emergency department staff.*


## 2. Materials and Methods

### 2.1. Sample and Procedure

The cross-sectional study was conducted in the form of an online survey. Study participants were enrolled via the Deutsche Gesellschaft für Interdisziplinäre Notfall- und Akutmedizin [German Association for Interdisciplinary Emergency Medicine and Acute Care, DGINA]. The DGINA has more than 1000 members. They comprise doctors, including in senior positions, nursing staff and paramedics and non-medical professionals that work in the field of emergency medicine. All members received an e-mail invitation to participate in the online survey via the DGINA mailing list. The invitation contained an information flyer about the study objective and procedure, the conditions of participation and data management. The link to the online survey was also included in the e-mail. The members were asked to take part in the survey, as well as to forward the invitation to their colleagues in emergency medicine via the snowball principle.

The data was collected from all over Germany from September to December 2020. Study participants had to be at least 18 years of age, work in emergency departments of hospitals or in the emergency services and have direct contact with patients and/or their relatives as part of their work. They included doctors, nursing staff and emergency responders, as well as non-medical staff, such as receptionists.

### 2.2. Measures

The online survey included sociodemographic information in part A, while part B recorded information about violent incidents, the frequency and nature of the incidents, the target of the aggression and measures taken, the stress and physical and/or psychological impacts, as well as about support from supervisors and colleagues (based on SOAS-R and developed internally).

Part C included information about personal coping mechanisms, ability to work, changing roles and mental health. This information was collected using the Copenhagen Burnout Inventory (CBI), WHO-Five Well-being index (WHO-5), the RS-13 resilience questionnaire, Work ability Index (WAI) and Copenhagen Psychosocial Questionnaire (COPSOQ).

The last part of the survey collected information about the preparation provided by the workplace and support after violent incidents, such as preventive measures, de-escalation training, reporting systems and follow-up care. Senior staff received additional questions about the hospital structure, how violent incidents are recorded and about protective and preventive measures.

#### 2.2.1. Workplace Violence

The term violence used in the study corresponds with the ILO’s definition. Questions addressed incidents of verbal or physical violence during the last 12 months by patients or their relatives regarding the time before the coronavirus pandemic by frequency, ranging from never, once a year, once a quarter, once a month, once a week to daily. Due to the restrictions on persons accompanying patients to the emergency department because of the coronavirus pandemic, questions were also asked as to whether violent incidents had changed in any way as a result.

#### 2.2.2. Burnout

In this study, the risk of burnout was determined using the CBI, a public domain questionnaire measuring the degree of physical and psychological fatigue experienced in three sub-dimensions of burnout: personal, work-related, and client-related burnout. The focus in our study was on work-related and client-related burnout scales.

According to Kristensen et al. [34], work-related burnout is defined as “the scale of physical and psychological fatigue perceived by a person in connection with their work”. The work-related burnout scale helps to identify people who are tired and attribute this tiredness to work-related factors and not to other factors, such as health problems or family responsibilities. Client-related or patient-related burnout is defined as “the scale of physical and psychological fatigue perceived by a person in connection with their work with clients”.

On a five-point Likert scale extending from “always or to a large extent” to “never/almost never”, the employees were asked to estimate the extent of their exhaustion. The information collected via the Likert scale was converted for each item into corresponding scores, from severe exhaustion/stress (“always”, 100 points) and the three intermediate levels 75 points, 50 points and 25 points to no exhaustion/stress (“never/almost never”, 0 points). The items on both CBI sub-scales were calculated to reveal individual overall scores. With a total score of 50 points or more (maximum possible: 100 points), the respondent is classified as being at moderate risk of burnout, while scores of 75 or more put them in the high-risk category [35].

#### 2.2.3. Resilience

In this study, resilience is measured using the RS-13 resilience questionnaire [36]. Resilience refers to psychological resilience and describes the phenomenon that some people remain healthy despite enormous pressures and health risks and recover better from disorders and diseases than others, and it can therefore be understood as a positive counterpart to vulnerability [36]. The RS-13 comprises 13 items formulated as statements, such as “If I have plans, I pursue them”. The items are placed on a seven-point Likert scale and assigned scores from one to seven. To evaluate the result, the scores were added up. In order to interpret the characteristics of RS-13, the scores were categorised as low, moderate and high (see Table 1).

#### 2.2.4. Preparation by the Workplace and Perceived Stress

Measures of perceived stress and feelings of being prepared for violent incidents by the workplace were obtained using visual, ten-point analogue scales. For the ordinal regression models, these were summarised as insufficient/low (1–3), moderate (4–7) and good/high (8–10) [1].

### 2.3. Statistical Analyses

Categorical variables are given as absolute values with associated proportion values. Continuous variables were stated with means and standard deviations (±SD) or 95% confidence intervals (95% CI) where it was deemed appropriate. Missing values are indicated in the tables.

The investigation of risk factors, particularly experiences of verbal and physical violence, on exhaustion and stress, was carried out using ordinal regression models. Available variables that could also influence the result, including potentially protective effects, were also included in the models. To do this, proportional odds models were used for which the proportional odds assumption was tested in advance using suitable chi-squared tests. Odds ratios with the relevant 95% CI were calculated in order to quantify the effect estimates. *P*-values below 0.05 were deemed statistically significant. SPSS version 27 (SPSS Inc., Armonk, NY, USA) and the statistical analysis program R in version 4.1.2 (R Foundation for Statistical Computing, Vienna, Austria) were used for the statistical analyses and graphical presentations.

#### Ethical Considerations

The data protection concept was developed in consultation with the data protection officer of the Institution for Statutory Accident Insurance and Prevention in the Healthcare and Welfare Services (BGW); the involvement of an ethics committee is not envisaged in the case of anonymous data collection. The principles of the declaration of Helsinki are followed.

## 3. Results

### 3.1. Descriptive Statistics

A total of 349 employees (doctors, nursing staff and emergency responders) from German emergency departments took part in the survey. Of these, 115 held senior positions (33%). The description of the study population can be found in Table 2.

A total 87% of employees reported that they had experienced physical violence from patients in the past twelve months, and 64% stated that they had experienced physical violence from relatives. Ninety-seven percent of respondents had been confronted by verbal abuse from patients, and 94% by verbal abuse from relatives. Regarding the question of whether the frequency of violent incidents had changed during the pandemic and the related access restrictions, 28.4% said that the number of violent incidents had decreased, 45.0% said that they had remained the same and 25.8% said that they had increased. More than half of the respondents (58%) stated that the violent incidents and aggression had affected their work. In terms of the emotional impact, 73% of employees stated that they felt annoyed, and 53% stated that they felt angry. Forty percent felt helpless and 26% reported experiencing anxiety (no table). The institutions do offer some measures regarding how to handle violence. For example, responses included guidelines on dealing with difficult patients (27%), case discussions and supervision (24%) and technical alarm systems (31%). Twenty-four percent of respondents stated that they were unaware of any measures being available. Overall, 9.7% of respondents felt that their workplaces had prepared them well for such incidents. Support after violent incidents was primarily provided by colleagues (73.9%). Of the 115 senior staff, 45 (39%) stated that their management took a clear stance against violence.

#### 3.1.1. Resilience

The question of resilience potentially acting as a buffer for burnout risk revealed that 44% of those answering the questionnaire had high resilience, 23% had moderate resilience and 29% of respondents had low resilience levels (Table 2).

#### 3.1.2. Burnout and Perceived Stress

Forty-two percent of respondents had a moderate to high work-related burnout risk. For patient-related burnout, 17% of respondents had a moderate to high burnout risk (Figure 1).

A total of 22.6% of employees stated that they felt under high stress as a result of the violence experienced. Table 3 shows the distribution of both burnout scales and perceived stress, broken down by the characteristics of the study group. Women showed higher average scores in the burnout scales and in terms of perceived stress. Employees in publicly-owned institutions also reported higher averages in the three scales. The more frequently verbal abuse was experienced from patients or relatives, the higher the average scores on the scales. However, where employees felt well prepared for potential violent incidents, all three scales showed the lowest average values.

### 3.2. Regression Analyses

The results of the ordinal regressions for both burnout scales show that experiences of both physical and verbal abuse had higher odds ratios but that these were not statistically significant (Figure 2 and Figure 3). Therefore, the hypotheses (1–4a and 1–4b) that more frequent experiences of violence from patients or relatives are associated with a higher burnout risk could not be confirmed.

Where employees feel well prepared by their workplace for potential violent incidents, there was a negative correlation with work-related burnout (OR 0.53, 95% CI 0.39–0.71) and patient-related burnout (OR 0.53, 95% CI 0.33–0.86), which confirms hypotheses 5a and 5b. High resilience values were also shown to be a significant protective factor for both burnout scales, thus confirming hypotheses 6a and 6b.

Based on the results from the ordinal regression, there was no statistically significant correlation between patient violence and high perceived stress (H1c and H3c). The hypotheses (H2c and H4c) that verbal and physical violence from relatives posed a risk factor (OR 1.69, 95% CI 1.13–2.53 and OR 1.61, 95% CI 1.08–2.39) for high levels of perceived stress were, however, confirmed on the basis of these results. The hypothesis (H5c) that good preparation by the institutions has a protective effect (OR 0.77, 95% CI 0.63–0.94) was also confirmed (Figure 4).

## 4. Discussion

Emergency department staff are frequently exposed to verbal abuse and physical violence from patients and their relatives. This has already been well documented in numerous studies [3,12,17,18]. This study also found high prevalence rates of both physical violence and verbal abuse. The study distinguished between violence from patients and from their relatives. The proportion of violence from relatives was recorded as being very high, where 64.4% of respondents had been affected by physical violence from relatives, and the proportion having experienced verbal abuse was 94%, which was not much lower than the figure for verbal abuse from patients. A study on the frequency of violent incidents in emergency departments in Oman came to a similar conclusion. A total of 87.4% of respondents stated that they had been the victim of a violent attack in the past 12 months. In 84% of cases, this took the form of verbal abuse, while 18.4% suffered a physical attack. It was established that the majority of attacks were perpetrated by relatives and visitors to the emergency department [37]. Murray et al. found in a systematic review of 104 articles that in the studies that measured the prevalence throughout an entire career, between 57% and 93% of emergency response staff reported having experienced verbal abuse and/or physical violence at least once in their career [12]. Another review of the prevalence of physical violence against health workers found a pooled annual prevalence of physical attacks by patients of 19.3% (95% CI 16.5–22.5%) [3].

Aggression by patients and/or their relatives towards healthcare workers seemed to be an increasingly prevalent stress factor in the day-to-day work environment. This study investigated the correlations between violent experiences by employees in German emergency departments, both of verbal and physical violence from patients or their relatives, and a possible burnout risk and high perceived stress. In particular, there was a correlation between aggression from relatives and high perceived stress, which indicates impaired mental health. With regard to a correlation between violent experiences and a burnout risk, while there were higher odds ratios both for physical and for verbal violence from patients and their relatives, these odds ratios were not statistically significant.

The extent to which violent experiences in the workplace affect the potential risk of burnout has been investigated in several studies. In the study on violent attacks and burnout in emergency departments in Chile, employees were severely affected by violent incidents. 71% of the 565 respondents had been the victim of violence in the past 12 months. The burnout level was measured at 10.5%. Being a victim of a violent attack was highly associated with emotional exhaustion (OR 1.7, 95% CI 1.1–2.8) and depersonalisation (OR 2.0, 95% CI 1.3–3.3) [28]. In a study from Palestine, experiencing physical violence was significantly associated with a high degree of burnout (OR 2.0, 95% CI 1.12–3.6, *p* = 0.002). However, no association between verbal abuse and burnout was proven (OR 1.8, 95% CI 0.87–3.39, *p* = 0.115) [38]. A study from the US, however, showed that exposure to verbal abuse can have a significant impact on burnout, job satisfaction and secondary stress among emergency department staff [22]. One German study investigated the correlation between violent experiences and the development of burnout in the emergency services. The sample size (*n* = 358) did not show higher burnout rates but 97.5% reported that they had already been insulted or spat at during the course of their work. The experience of feeling threatened was found to be a significant predictor of emotional exhaustion. The authors concluded that specific violent experiences by the emergency services could be associated with burnout symptoms [39]. These results show that violent experiences can impact the health of employees and may pose a risk for burnout.

Furthermore, where employees felt well prepared by their workplace for potential attacks, this was negatively associated with burnout risk and high perceived stress. In a study on violent attacks against healthcare workers from 2019, preparation by the workplace for potential attacks was also shown to be a significant protective factor against high perceived stress [1]. High resilience is also in negative correlation to burnout in our study, thus supporting the results of various studies [23,33] that showed that lower or moderate resilience was associated with higher burnout values.

This puts the focus on preventive measures. The results of our study increase our understanding of how to handle aggression by showing that effective preparation by the institution can help to reduce the potential negative impact of violence and aggression on employees’ mental health.

### 4.1. Limitations and Future Directions

Some of the limitations of our study must be mentioned. Our study took the form of an anonymous online survey. As a result of the cross-sectional design, no causal relationships could be established. The participants were recruited via a link to the survey, which was distributed by DGINA via a snowball system. As a result, it was not possible to calculate response rates. This may have led to a selection bias. This enabled the survey to be carried out throughout Germany, however. A total of 349 employees took part in the survey.

In our cross-sectional study, the violent incidents were recorded retrospectively over a period of 12 months for the time before the coronavirus pandemic, which means that a recall bias cannot be precluded. The special conditions resulting from the coronavirus pandemic may also have affected the results. At the time of the survey, no accompanying persons were allowed to enter the emergency department. We attempted to take this into account in the survey. Thus, we explicitly asked about violent incidents prior to the coronavirus pandemic. In an attempt to reduce bias, an additional question was also included regarding whether the violent incidents had changed during the pandemic.

### 4.2. Practical Implications

One central consideration of this study is the availability of resources because they can help employees to stay healthy despite experiencing violent incidents. This study showed that preparation by the institution and good resilience were significant protective factors with regard to burnout risk and perceived stress. However, only 10% of respondents found that their institution had prepared them effectively for such situations. In this context, knowledge sharing and strategies for the prevention of violence by the institution’s management play a very important role in minimising the risk of burnout and perceived stress among employees. The fact that only few employees felt well prepared clearly shows that many management staff do not perceive this task to be part of their remit, or are overwhelmed by the concept. One reason may be that the management has not adopted a clear stance against violence in the workplace. Only 39% of senior employees stated that the management had taken a clear stance against violence. If the topic is not taken seriously by the highest levels of management and a clear stance against violence and aggression is not taken, the lower management levels and employees will find it hard to create preventive strategies or develop an open culture of discussion. Supervisors and management should therefore be trained in promoting the job-related resources of their staff by actively supporting their employees, for example, in developing specific supportive measures and preventive strategies, such as resilience training and remaining approachable in terms of listening to staff issues (emotional support). This makes violence prevention a key management task, which contributes significantly towards ensuring a safe workplace. This has also been shown by a Swiss study. The study investigated the views of the senior ward staff and nurses with regard to the ability of nursing teams to deal with aggression from patients and visitors in clinical practice, along with the related challenges. Senior nursing staff are seen as key people for helping their teams develop efficient ways to handle aggression from patients and visitors. Their perception of patient and visitor aggression in clinical practice has, however, rarely been investigated and issues relating to team management in this context have not been sufficiently analysed. The analysis of the focus groups revealed relevant topics: a culture of organisational safety and cooperation, factors within the team, such as team culture, nursing aggression and general management principles and implications for the management of nursing staff. The authors came to the conclusion that handling patient and visitor aggression is a challenge and major management topic and that the ability of a team to prevent, de-escalate and follow-up after violent incidents depends on the management staff. However, the management had rarely been trained and require more support in order to put their teams in a position to be able to handle aggression from patients and visitors more effectively. The implementation of guidelines and principles within the institution should be made a higher priority [40].

Even if this study indicates that support from supervisors can be an effective resource, it is also important to indicate that the institution’s main aim should be to prevent the occurrence of aggression and violence from patients and relatives. This should be done across several levels of prevention and should involve technical and environmental, organisational and behavioural measures. According to the Guidelines for the Prevention of Workplace Violence in Healthcare, this could include technical measures such as controlled access to the emergency department. Due to the coronavirus pandemic, patients have not been accompanied by relatives. This led to lower visitor numbers in the emergency departments and was seen as a positive step by employees. Good lighting, comfortable and spacious waiting areas, alarm systems and surveillance cameras and security personnel could also help to improve safety. At an organisational level, adequate staffing, open communication about violence and aggression, optimised workflows and good documentation are important elements. At the behavioural level, this includes training workshops and behavioural training for employees and supervisors that include strategies and processes such as de-escalation training and self-defence techniques. These can help to give employees the feeling that they are able to handle critical situations safely and competently. The results of a systematic review from 2021 support this. They show that the studies reviewed generally observed a positive effect of behaviour-oriented and multidimensional interventions on reducing violent attacks by patients towards emergency department staff. They were also able to improve the ability of staff to handle violent situations [32].

## 5. Conclusions

Verbal abuse and physical violence and aggression targeting employees in German emergency departments are very common and may pose a health risk to those affected. Aggression may be perpetrated by patients or by their relatives. Violent experiences in the workplace may increase the risk of burnout and perceived stress by those affected. Violence by relatives had a particularly negative effect on perceived stress. High resilience or effective preparation of employees for potential attacks were shown to have a protective effect with regard to the burnout risk and perceived stress. Management staff play a major role in preventing violence and its impact on employees. The senior management should take a clear stance against violence in the workplace and strengthen the resources of their staff and managerial staff.

## Figures and Tables

**Figure 1 ijerph-19-04945-f001:**
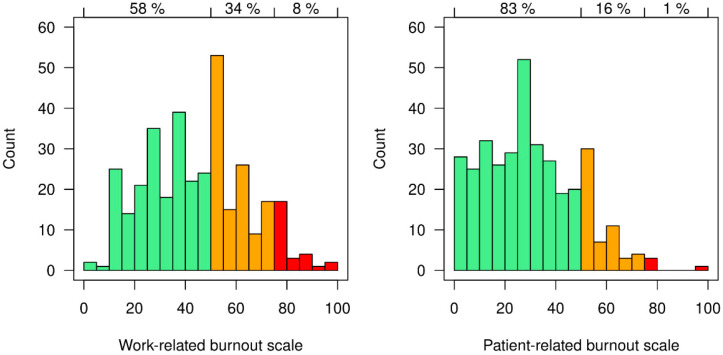
Data on the Copenhagen Burnout Inventory (CBI) scales for work-related and patient-related burnout risk (Green = low; orange = moderate; red = high).

**Figure 2 ijerph-19-04945-f002:**
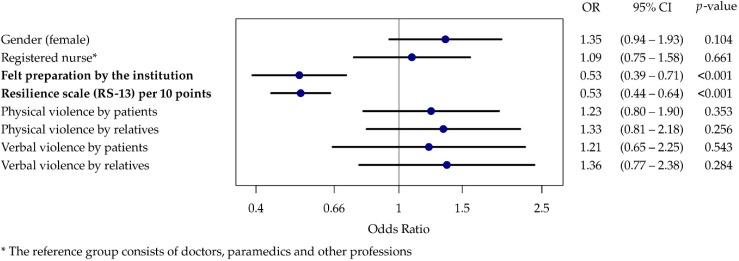
Forest plot of the ordinal regression results for work-related burnout.

**Figure 3 ijerph-19-04945-f003:**
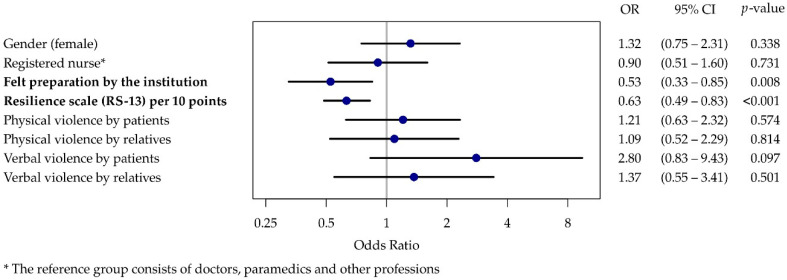
Forest plot of the ordinal regression results for patient-related burnout.

**Figure 4 ijerph-19-04945-f004:**
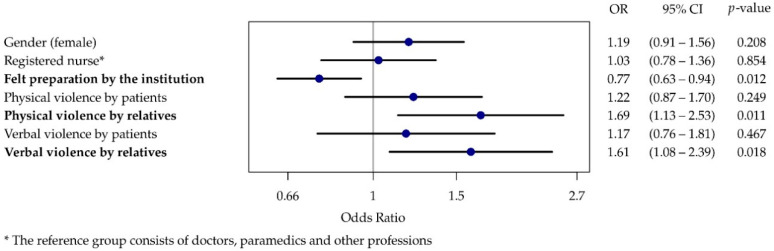
Forest plot of the ordinal regression results for high perceived stress due to violence experienced.

**Table 1 ijerph-19-04945-t001:** Qualitative categorisation of the Copenhagen Burnout Inventory (CBI) and the RS-13 resilience questionnaire.

	Low	Moderate	High
CBI, work-related	0–<50	50–<75	->75–100
CBI, patient-related	0–<50	50–<75	->75–100
RS-13	13–66	67–72	73–91

**Table 2 ijerph-19-04945-t002:** Description of the study population (*n* = 349).

Variables	*N*	%
**Age**		
- 20 to 29 years	100	28.7
- 30 to 39 years	98	28.1
- 40 to 49 years	80	22.9
- 50 to 59 years	59	16.9
- ≥60 years	11	3.2
- N/A	1	
**Gender**		
- Female	202	57.9
- Male	146	41.8
- N/A	1	
**Occupation**		
- Doctor	85	24.3
- Nurse	208	59.6
- Paramedic	25	7.2
- Other	31	8.9
**Type of institution**		
- Private	63	18.1
- Public	183	52.4
- Independent	102	29.2
- N/A	1	
**Experiences of physical violence from patients**		
- Yes	305	87.4
- No	42	12.0
- N/A	2	
**Experiences of physical violence from relatives**		
- Yes	225	64.5
- No	121	34.7
- N/A	3	
**Experiences of verbal abuse from patients**		
- Yes	339	97.1
- No	8	2.3
- N/A	2	
**Experiences of verbal abuse from relatives**		
- Yes	329	94.3
- No	18	5.2
- N/A	2	
**Perceived stress due to violence experienced**		
- Low	68	19.5
- Moderate	195	55.9
- High	79	22.6
- N/A	7	
**How effective were your workplace’s measures preparing you for violent incidents?**		
- Insufficient	159	45.6
- Moderate	154	44.1
- Good	34	9.7
- N/A	2	
**Received support following incidents**		
-Received support from supervisors	86	24.6
- Received support from colleagues	258	73.9
- Received support from persons outside the workplace	96	27.5
- No support received	71	20.3
- N/A	4	
**Resilience**		
- Low	101	28.9
- Moderate	80	22.9
- High	155	44.4
- N/A	13	

**Table 3 ijerph-19-04945-t003:** Work-related and patient-related burnout and perceived stress according to general characteristics (mean ± SD).

Variable	CBI, Work-Related	CBI, Patient-Related	Perceived Stress
**Gender**			
- Male	40.4 ± 19.0	27.7 ± 17.1	5.4 ± 2.5
- Female	46.4 ± 20.2	31.3 ± 18.8	5.9 ± 2.2
**Age**			
- 20 to 29 years	45.1 ± 21.1	32.7 ± 17.0	5.8 ± 2.4
- 30 to 39 years	45.5 ± 21.2	30.1 ± 20.1	5.6 ± 2.3
- 40 to 49 years	43.2 ± 17.6	29.8 ± 18.6	5.8 ± 2.6
- 50 to 59 years	40.5 ± 18.5	25.1 ± 16.0	5.5 ± 2.2
- ≥60 years	40.9 ± 19.8	26.3 ± 14.5	5.1 ± 1.7
**Occupation**			
- Doctor	44.1 ± 19.1	26.8 ± 18.2	5.4 ± 2.6
- Nurse	44.8 ± 19.5	31.0 ± 17.3	5.8 ± 2.1
- Paramedic	30.2 ± 16.6	22.9 ± 16.2	4.2 ± 2.4
- Other	48.3 ± 23.2	35.0 ± 22.4	6.2 ± 2.5
**Type of institution**			
- Private	47.6 ± 22.6	30.8 ± 18.1	6.3 ± 2.5
- Public	45.2 ± 19.3	31.1 ± 18.2	5.6 ± 2.3
- Independent	39.5 ± 18.5	26.8 ± 17.5	5.4 ± 2.3
**Time working in emergency department**			
- 0 to 5 years	46.7 ± 21.1	32.6 ± 16.5	5.8 ± 2.3
- 6 to 10 years	45.6 ± 18.7	31.0 ± 18.6	5.7 ± 2.3
- 11 to 15 years	45.8 ± 21.8	30.7 ± 20.7	5.8 ± 2.2
- Longer than 15 years	38.7 ± 17.7	25.3 ± 17.2	5.3 ± 2.5
**Verbal abuse from patients**			
- Never	30.8 ± 20.5	16.2 ± 9.8	3.5 ± 3.0
- Once a year	36.6 ± 19.5	25.5 ± 17.9	4.8 ± 2.7
- Once a quarter	40.8 ± 19.2	24.5 ± 13.7	4.8 ± 2.2
- Once a month	39.9 ± 20.5	24.7 ± 15.8	5.2 ± 2.2
- Once a week	44.9 ± 17.9	32.4 ± 18.1	6.0 ± 2.3
- Daily	52.0 ± 20.5	37.5 ± 21.0	6.6 ± 2.0
**Verbal abuse from relatives**			
- Never	31.2 ± 15.7	16.2 ± 12.7	2.6 ± 1.8
- Once a year	38.5 ± 18.0	22.7 ± 15.0	4.7 ± 2.6
- Once a quarter	41.6 ± 18.7	26.6 ± 15.0	5.1 ± 2.2
- Once a month	41.2 ± 20.6	26.8 ± 17.0	5.7 ± 2.2
- Once a week	47.2 ± 19.3	32.2 ± 17.2	6.0 ± 2.2
- Daily	50.6 ± 20.2	39.3 ± 21.4	6.7 ± 1.9
**Physical violence from patients**			
- Never	36.5 ± 20.6	22.3 ± 14.0	4.4 ± 2.5
- Once a year	41.3 ± 18.1	27.4 ± 15.3	5.4 ± 2.3
- Once a quarter	43.7 ± 20.1	29.3 ± 18.6	5.6 ± 2.3
- Once a month	46.1 ± 19.5	31.1 ± 19.6	6.0 ± 2.2
- Once a week	52.8 ± 19.0	38.6 ± 18.7	6.8 ± 1.8
- Daily	40.0 ± 22.1	35.8 ± 17.1	7.0 ± 2.7
**Physical violence from relatives**			
- Never	39.6 ± 20.1	25.7 ± 16.3	4.8 ± 2.3
- Once a year	46.0 ± 18.6	30.8 ± 18.6	5.8 ± 2.2
- Once a quarter	43.2 ± 19.3	30.6 ± 17.7	6.0 ± 2.2
- Once a month	50.7 ± 21.0	33.0 ± 21.8	6.7 ± 2.2
- Once a week	49.8 ± 20.0	38.5 ± 16.6	7.0 ± 1.4
- Daily	30.4 ± 17.7	29.2 ± 11.8	5.5 ± 3.5
**Preparation by institution**			
- Good	35.7 ± 15.6	24.2 ± 16.2	4.6 ± 2.3
- Average	41.3 ± 18.8	27.3 ± 16.3	5.7 ± 2.2
- Insufficient	48.3 ± 20.9	33.3 ± 19.6	5.9 ± 2.4

## Data Availability

The data are available upon request from the corresponding author.

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
