# Peer review of "Risk of Burnout among Emergency Department Staff as a Result of Violence and Aggression from Patients and Their Relatives"

_ijerph, 2022, doi:10.3390/ijerph19094945_

Round 1
Reviewer 1 Report
The study generate data on the frequency of violence by patients and relatives and to gain a better understanding of the connection between experienced aggression, a possible risk of burnout and a high sense of stress. The current study has merits and contributes to the existing knowledge in its current format. However, it needs to incorporate some changes. I recommend the author(s) make the suggested changes indicated below and re-submit the article for consideration:
- Although the research mentioned the purpose of the research, there were no clear research questions. It is suggested that additional explanation should be provided. For example: What are the “practical” issues that need to be resolved? Therefore, the motivation for the paper is unclear. I'm not sure what problem the paper is meant to address. It does outline a situation that the authors would like to explore, but there is no real research gap identified or contributions that might arise from studying this problem. As a result, it is very uncertain to the reader what they're going to find out after they make it to the end of this paper.
- In order to make a clear contribution to the literature, I urge the authors to make a greater contribution that is grounded in a well-though “theoretical background”: The current description of the research background is still simple, and the literature is old, just 3 references are 2021, it should be updated.
- The research hypotheses seem to be rather simple and easily answered through some reading of the literature. Please enhance the arguments and justifications for each hypothesis, and strengthen the research hypothesis derivation process.
- Table 2 is not easy to read, the authors may consider modified to four fields such as variables, items, N, % . The Cumulative percentages should add up to 100.
- The authors may consider move the “4.1. Limitations and future directions” section to after “5. Conclusions”.
Author Response
Response Reviewer 1
Answers to the Reviewers:
Reviewer 1: Comments and Suggestions for Authors
The study generate data on the frequency of violence by patients and relatives and to gain a better understanding of the connection between experienced aggression, a possible risk of burnout and a high sense of stress. The current study has merits and contributes to the existing knowledge in its current format. However, it needs to incorporate some changes. I recommend the author(s) make the suggested changes indicated below and re-submit the article for consideration:
- Although the research mentioned the purpose of the research, there were no clear research questions. It is suggested that additional explanation should be provided. For example: What are the “practical” issues that need to be resolved? Therefore, the motivation for the paper is unclear. I'm not sure what problem the paper is meant to address. It does outline a situation that the authors would like to explore, but there is no real research gap identified or contributions that might arise from studying this problem. As a result, it is very uncertain to the reader what they're going to find out after they make it to the end of this paper.
Answer to the Reviewer: Thanks for your suggestions. We revised the introduction section and our research questions more clearly. Also we addresed the practical issue that need to be resolved.
It is true that there are numerous studies on the risk of burnout among employees and this has been well researched. However, there are differences between occupational groups and departments. Little research has been done on the subject of emergency room workers, and there is almost no literature on the subject in German-speaking Europe. The focus here should also be on the influence of experiences of violence on possible burnout and feelings of stress. In this context, the results of our studies provide important insights to improve the prevention of violent experiences.
2. In order to make a clear contribution to the literature, I urge the authors to make a greater contribution that is grounded in a well-though “theoretical background”: The current description of the research background is still simple, and the literature is old, just 3 references are 2021, it should be updated.
Answer to the Reviewer: We revised this point and added more new references.
3. The research hypotheses seem to be rather simple and easily answered through some reading of the literature. Please enhance the arguments and justifications for each hypothesis, and strengthen the research hypothesis derivation process.
Answer to the Reviewer: See the points above we revised the background section.
4. Table 2 is not easy to read, the authors may consider modified to four fields such as variables, items, N, %. The Cumulative percentages should add up to 100.
Answer to the Reviewer: Thanks, we revised the Table 2 and now the cumulative percentages add up to 100.
5. The authors may consider move the “4.1. Limitations and future directions” section to after “5. Conclusions”.
Answer to the Reviewer: Thanks for your comment. But we used the template from the journal with the prescribed layout.

Reviewer 2 Report
Review of “Risk of burnout among emergency department staff as a result of violence and aggression from patients and their relatives”
This manuscript on the topic of burnout among emergency department staff was interesting and adds to the burnout and stress literature among healthcare professionals. My major recommendations are below:
- The authors use hypotheses for the study. Since the understanding of the topic is still evolving and being researched, should these have been research questions? This decision will be left most likely to the editors and authors.
- As for the measures, the burnout scale focused on emotional exhaustion. However, burnout includes other categories as well, including depersonalization and personal accomplishment measures (or related terms such as cynicism and professional efficacy, respectively). Were these not assessed? If not, why? What was the certain burnout scale used?
- It would be interesting to see the results by test instrument. Some are shown in Table 1, but others could be included for greater clarity.
- For the regressions were the different types of employees assessed? It appears nurses were used as there is a category called examined nurse. This needs to be explained.
- It appears that the same charts on repeated in Figure 2.
- The methods show that the respondents discussed events over a 12-month period but were told to consider events before the pandemic? Were the respondents only responding with pre-COVID data for all responses? This is not clear.
- Acronyms need to be clearly defined throughout the paper.
- Arabic numbers with percentages should not be used to start sentences. These numbers that start sentences should be written out. See for example line 239 on page 7. It should read Twenty-four percent….
- In several of the charts the column results need to be lined up better with the variables.
Author Response
- The research hypotheses seem to be rather simple and easily answered through some reading of the literature. Please enhance the arguments and justifications for each hypothesis, and strengthen the research hypothesis derivation process.
Review of “Risk of burnout among emergency department staff as a result of violence and aggression from patients and their relatives”
This manuscript on the topic of burnout among emergency department staff was interesting and adds to the burnout and stress literature among healthcare professionals. My major recommendations are below:
2. The authors use hypotheses for the study. Since the understanding of the topic is still evolving and being researched, should these have been research questions? This decision will be left most likely to the editors and authors.
Answer to the Reviewer: Thank you for pointing this out, we have revised and clarified the research questions accordingly.
We do not think that there is enough research on this issue among emergency workers. There are indeed some studies on the burnout risk of HWC. Often, however, they do not examine the connection between violence experienced at the workplace and the risk of burnout. In addition, there are differences in the occupational groups as well as in the different settings such as hospitals, care of the elderly, etc. However, there are rather few studies on employees in emergency rooms and none in German-speaking Europe. Initial studies on this topic also showed deficits in prevention measures in this sensitive area of emergency rooms. Therefore, it is important in this context to show what influence the experience of violence can have on the health of employees and what factors contribute to reducing the risk in order to make the problem clear to decision-makers in emergency rooms. Our results contribute to this.
3. As for the measures, the burnout scale focused on emotional exhaustion. However, burnout includes other categories as well, including depersonalization and personal accomplishment measures (or related terms such as cynicism and professional efficacy, respectively). Were these not assessed? If not, why? What was the certain burnout scale used?
Answer to the Reviewer:
4. It would be interesting to see the results by test instrument. Some are shown in Table 1, but others could be included for greater clarity.
Answer to the Reviewer: Thanks for your suggestion. Thank you for your comment. The descriptive results for the burnout scales are shown in Figure 1. We have added the results for the stress perception. In our opinion, all results of the test instruments are now described
5. For the regressions were the different types of employees assessed? It appears nurses were used as there is a category called examined nurse. This needs to be explained.
Answer to the Reviewer: Thank you for your comment. It is true that we have adjusted for the occupations. We have now explained this in a footnote and made it easier to understand.
It appears that the same charts on repeated in Figure 2.
Answer to the Reviewer: Thank you for the information. An error had occurred in the revision mode. We have corrected this.
6. The methods show that the respondents discussed events over a 12-month period but were told to consider events before the pandemic? Were the respondents only responding with pre-COVID data for all responses? This is not clear.
Answer to the Reviewer: Thank you very much for the comment. This point was not clearly expressed. The question was about experiences of violence in the last 12 months before the Corona pandemic. Since the survey was taken during the pandemic and there were changes in the access restrictions for accompanying persons in the process of the survey, an additional question was included to find out whether there had been any changes in the incidence of violence as a result. We revised this point. This is now described in more detail in the text
7. Acronyms need to be clearly defined throughout the paper.
Answer to the Reviewer: Thank you fort he comment. We changed this accordingly.
8. Arabic numbers with percentages should not be used to start sentences. These numbers that start sentences should be written out. See for example line 239 on page 7. It should read Twenty-four percent….
Answer to the Reviewer: Thanks for the comment, we have adjusted and changed this accordingly.
9. In several of the charts the column results need to be lined up better with the variables.
Answer to the Reviewer: Thanks for the comment, we have adjusted and changed this.

Round 2
Reviewer 1 Report
Dear Authors,
Thank you for the revisions that you have made. I am happy about all, although the paper was not revised completely, but in its actual form can be considered for publication.